# Sleep Quality in Women with Premenstrual Syndrome Is Associated with Metabolic Syndrome-Related Variables

**DOI:** 10.3390/healthcare11101492

**Published:** 2023-05-20

**Authors:** Hyejin Chun, Miae Doo

**Affiliations:** 1Department of Family Medicine, Ewha Womans University College of Medicine, Seoul 07804, Republic of Korea; fmewha@naver.com; 2Department of Food and Nutrition, Kunsan National University, Gunsan 54150, Republic of Korea

**Keywords:** dietary habits, metabolic syndrome, sleep quality, premenstrual syndrome

## Abstract

In this study, we examined whether metabolic syndrome (MetS)-related variables are simultaneously affected by sleep quality, premenstrual syndrome (PMS) and dietary consumption. In this cross-sectional study, data for 307 premenopausal women were available. The results showed that women experiencing PMS had significantly lower sleep quality and were more depressed and anxious (*p* < 0.001 for all). After the subjects were divided into groups according to PMS, the effect of sleep quality on MetS-related variables or MetS components significantly differed; only among women who experienced PMS were poor sleepers significantly higher in waist circumference (*p* = 0.018) and diastolic blood pressure (*p* = 0.012) than good sleepers. Among the MetS components, abdominal obesity in women with poor sleep quality was approximately three (16.9% vs. 3.0%, *p*= 0.020) times more common than in those with good sleep quality. However, these findings were not observed among those who did not experience PMS. Poor sleepers among women experiencing PMS consumed 2.8 times more alcoholic drinks than good sleepers (*p* = 0.006). The MetS-related variables in Korean women experiencing PMS are associated with sleep quality, and these associations may be modified by dietary habits.

## 1. Introduction

Premenstrual syndrome (PMS) is characterized by a combination of somatic and psychological symptoms that occur during the luteal phase of the menstrual period, with symptoms ending after menstruation [1]. The symptoms of PMS include depression, anxiety, irritability, edema, social isolation, abdominal cramps, appetite changes and sleep problems [2]. The symptoms vary among different individuals; many women experience their periods with mild symptoms or without any symptoms at all. Some patients suffer severe PMS symptoms, which is indicative of a condition called premenstrual dysphoric disorder (PMDD) [3].

Many studies have reported that PMS in women is associated with health-related factors. One of the main causes of sleep problems in women of childbearing age is PMS, which results in insomnia, short sleep duration, daytime sleepiness and frequent nighttime awakenings [4,5]. A recent meta-analysis reported that alcohol consumption has a harmful effect on PMS [6]. Indeed, dietary consumption is associated with the presence or severity of PMS. For example, high consumption of fat, fast foods, fried foods, red meats and soft drinks or low consumption of fruits and vegetables were associated with the presence and increased severity of PMS [7,8,9]. Moreover, high consumption of fat, especially saturated fats, was positively associated with the symptoms of PMS [10]. Women who consume a high-fat diet have increased levels of estrogen, which results in the presence of symptoms such as breast swelling and tenderness, bloating, nodularity and water retention [11].

Metabolic syndrome (MetS) is a complex disorder that is defined by a cluster of factors: central obesity, hypertriglyceridemia, low HDL cholesterol, increased blood pressure and impaired blood glucose. MetS and its components are associated with an increased risk of cardiovascular disease, which is one of the leading causes of morbidity and mortality worldwide [12]. Risk factors that predispose individuals to MetS are components of unhealthy lifestyles, including dietary patterns, drinking, smoking, sleep behavior and physical activity, as well as genetic factors [13]. Thus, the management of MetS requires healthy lifestyle modifications.

Although there is no consensus to date that the presence of PMS directly affects MetS, one study has demonstrated that Iranian women with PMS had a significantly increased risk of MetS [14]. Additionally, some studies have reported that women who experienced obesity showed higher concentrations of TGs, lower HDL cholesterol [14], increased abdominal obesity [15] and increased risk for hypertension [16], which are components of MetS.

Therefore, we hypothesized that women who experienced PMS might have different MetS-related variables, since PMS affects sleep quality. Accordingly, the purpose of our study was to identify how MetS-related variables are affected by different levels of sleep quality and the presence of PMS among Korean women. In addition, dietary consumption was investigated according to different levels of sleep quality and the presence of PMS.

## 2. Subjects and Methods

### 2.1. Study Design and Subject Selection

The subjects in this cross-sectional study were premenopausal women who visited the hospital for health check-ups from September 2020 to June 2021. Before participating in the survey, among women who wanted to participate voluntarily, we asked if they met the following inclusion criteria: (1) having a regular menstrual cycle; (2) not having pregnant or breastfeeding status; (3) not having depression or anxiety and (4) not having sleep disorders (Figure 1). A total of 341 subjects satisfied the criteria and were included. We conducted a comprehensive face-to-face interview and assessed the anthropometric and laboratory variables of the subjects. Through subsequent hormone tests, we excluded subjects who experienced menopause or irregular menstrual cycles due to polycystic ovary syndrome or prolactinoma (*n* = 11). Additionally, we excluded women taking antidepressants or anxiety medications (*n* = 4) and those taking sleep disorder medications (*n* = 7). Furthermore, subjects with missing or inadequate anthropometric and laboratory data for assessing MetS (*n* = 7) and those having implausible daily consumption (total energy consumption of ≤500 kcal or ≥3500 kcal) were excluded (*n* = 5). Finally, a total of 307 participants were included in the analyses. Informed consent was obtained from all participants, and the study was approved by the Institutional Review Board (IRB No. 1040117-201905-HR-004-02, IRB No. CHAMC 2020-07-005-002).

### 2.2. Data Collection

Data on the presence of PMS, sleep quality, general characteristics, metabolic syndrome-related variables and diet were collected from subjects. The data on the presence of PMS, sleep quality, general characteristics and dietary consumption were collected using a newly developed questionnaire. MetS-related variables were assessed using data from health examinations at the health care center. The premenopausal status of subjects was confirmed through the following hormone tests: serum estradiol, progesterone, sex hormone binding globulin (SHBG) and prolactin.

The questionnaire was composed to investigate data on socioeconomic variables (including age, educational level, occupation and marriage status), health-related variables (including current smoking, alcohol drinking, physical activity, as assessed by the international physical activity questionnaire, dietary supplement use, depression and anxiety), presence of PMS, sleep quality and dietary consumption.

PMS was assessed using the Menstrual Distress Questionnaire (MDQ) developed by Moos [17] and modified and supplemented by Kim [18]. Symptoms experienced at least 2 weeks before the onset of menstruation were included. It has 41 items and is composed of a 5-point scale, with higher scores indicating severe PMS. Using these media, the study subjects were divided into two groups: women who experienced PMS (*n* = 152) and those who did not experience PMS (*n* = 155).

Sleep quality was assessed using the Pittsburgh Sleep Quality Index (PSQI) [19]. The PSQI consists of 19 items and comprises seven sleep quality scores, including perceived sleep quality, sleep latency, sleep duration, sleep efficiency, sleep disturbances, use of sleeping medication and daytime dysfunction. The global PSQI score ranges from 0 to 21, with PSQI ≤ 5 indicating “good sleepers” and PSQI > 5 indicating “poor sleepers”.

Depression was assessed using the Center for Epidemiological Studies–Depression (CES-D) by Radloff [20]. It has 20 items, with higher scores indicating greater depressive symptoms. Anxiety was assessed using the State–Trait Anxiety Inventory [21]. It has 20 items for trait anxiety and state anxiety. Our study used ‘state anxiety’, which measures subjectively perceived tension and anxiety in specific situations and current mood states. Higher scores indicate greater anxiety.

Dietary consumption was determined with face-to-face interviews by trained dietitians using a single 24-h recall record method. The typical number of alcoholic drinks consumed was investigated. The dietary consumption data were analyzed using CAN Pro 5.0 (computer-aided nutritional analysis program for professionals, Seoul, Republic of Korea) software, a nutrient database developed by the Korean Nutrition Society. Then, they were converted to dietary nutrient density, which is the amount of nutrients contained in 1000 kcal of specific nutrient consumption. Additionally, the obtained food items were categorized into food groups based on common food groups classified in the Korean Nutrient Database [22].

MetS-related variables included waist circumference (WC) and BP (blood pressure), including systolic blood pressure (SBP) and diastolic blood pressure (DBP), FG (fasting glucose), TG (triglyceride) and HDL-C (high-density lipoprotein cholesterol). MetS was defined based on the US National Cholesterol Education Program/Adult Treatment Panel III (NCEP/ATP III) [23] criteria of ≥3 of 5 of the following risk factors: (1) abdominal obesity, indicated by a WC of ≥85 cm according to the Korean Society for the Study of Obesity [24]; (2) high triglyceridemia, indicated by TGs ≥150 mg/dL or receiving treatment for high TGs; (3) low HDL-C (<50 mg/dL) or receiving treatment for low HDL-C; (4) high BP (≥130/85 mmHg) or receiving treatment for high BP and (5) high FG (≥100 mg/dL) or under treatment for high blood glucose.

### 2.3. Statistical Analyses

Tests of normality for all continuous variables were carried out before the analyses, and skewed variables were logarithmically transformed. Data are presented as back-transformed means and 95% confidence intervals (CIs). To assess general characteristics, including socioeconomic variables and health-related variables according to the presence of PMS, independent *t* tests were used for continuous variables, and *Pearson*’s χ^2^ test or *Fisher*’s exact test was used for categorical variables. After subjects were divided according to the presence of PMS, a general linear model was used to confirm the effects of sleep quality on metabolic syndrome-related variables and dietary nutrient density adjusted for depression level and anxiety level. A multivariate logistic regression adjusting for depression and anxiety levels was analyzed for the prevalence of metabolic syndrome and its components by sleep quality and presence of PMS. Differences in food groups per 1000 kcal according to sleep quality and the presence of PMS were analyzed using a general linear model after adjusting for depression level and anxiety level. A *p* value of <0.05 was considered to indicate statistical significance. All statistical analyses were performed using SPSS (version 27.0; IBM Corp., Armonk, NY, USA) software for Windows.

## 3. Results

The general characteristics dichotomized by the presence of premenstrual syndrome are shown in Table 1. The average age and sleep quality score measured using the PSQI were 39.58 years and 5.38, respectively. No significant differences in socioeconomic variables such as age, educational level, occupation and marriage status were found according to the presence of PMS. Among health-related variables, sleep quality, depression level and anxiety level were significantly higher in subjects who experienced PMS than in subjects who did not experience PMS (*p* < 0.001 for all). However, current smoking, alcohol drinking, physical activity by IPAQ and dietary supplementary intake were not significantly different according to the presence of PMS.

After the subjects were divided into groups according to the presence of PMS, to identify whether sleep quality affected MetS-related variables, a general linear model was used after adjusting for depression level and anxiety level (Table 2). Within the group that experienced PMS, subjects with poor sleep quality had significantly higher WC (74.10 cm vs. 70.36 cm, *p* = 0.018) and DBP (72.99 mmHg vs. 67.95 mmHg, *p* = 0.012) than those with good sleep quality. However, among groups who did not experience PMS, no significant differences were observed in WC and DBP according to sleep quality. Regardless of the experience of PMS, SBP, FG, TG and HDL-C did not differ with sleep quality.

The prevalence of MetS and its components according to sleep quality and the presence of PMS are listed in Figure 2. After controlling for covariates, a multivariate logistic regression model was used for the differences in the prevalence of MetS and its components according to sleep quality and the presence of PMS. The influence of PMS differences on the prevalence of MetS and its components were observed. In other words, within the group of subjects who experienced PMS, subjects with poor sleep showed a higher trend in the prevalence of MetS and its components than those with good sleep, regardless of statistical significance. On the other hand, trends according to sleep quality were difficult to find among those who did not experience PMS. Among the MetS components in subjects with poor sleep quality with PMS, abdominal obesity was approximately five times more common than in those with good sleep quality and without PMS (16.9% vs. 3.0%, *p* = 0.020).

After being stratified into groups according to the presence of PMS, the consumption of nutrients and food groups according to sleep quality are represented in Table 3 and Table 4, respectively. Dietary nutrient density was calculated as the ratio of nutrients to calories consumed. There were no significant differences between all dietary nutrient densities and sleep quality, regardless of the presence of PMS. Differences in the consumption of food groups per 1000 kcal were observed between subgroups of sleep quality and the presence of PMS (Table 4). Among subjects who did not experience PMS, milk and dairy products were consumed approximately 2 times more often by good sleepers than by poor sleepers (57.56 g vs. 28.84 g, *p* = 0.020). Different trends were identified in subjects who experienced PMS. In other words, among subjects who experienced PMS, subjects with good sleep showed significantly higher consumption of eggs than those with poor sleep (29.39 g vs. 16.69 g, *p* = 0.028). However, alcoholic drinks were consumed less by good sleepers than by poor sleepers (24.9 g vs. 70.9 g, *p* = 0.006).

## 4. Discussion

The aim of this study was to examine whether sleep quality influenced MetS-related variables or dietary consumption in relation to the presence of PMS in a community-based cross-sectional study conducted on 307 Korean women. When stratified by the presence of PMS in a model that adjusted for depression level and anxiety level, the MetS-related variables and prevalence of MetS components according to sleep quality were significantly different. Additionally, the consumption of food groups showed significant differences in good and poor sleep quality after being divided into groups according to the presence of PMS.

The findings of our study are consistent with those of previous studies [4,5,25,26]. Women with PMS suffer more psychological abnormalities, such as symptoms of depression, anxiety, irritability and poor sleep quality. These are also some of the many symptoms that appear as PMS. Meers et al. reported that sleep deprivation and psychological abnormalities co-occurred, rather than existing as separate correlated factors associated with the severity of PMS and negative mood [26]. Although the cause of PMS has not been clearly defined as one of the explanatory factors, hormonal imbalance is associated with the presence or severity of PMS. A study of women with diagnosed PMDD reported that a sudden reproductive hormone (estradiol and progesterone) change was responsible for triggering negative mood or poor sleep [27]. Metabolic syndrome-related variables and the prevalence of MetS and its components according to the presence of PMS were not observed. Accordingly, we stratified the participants into groups according to the presence of PMS, and then a general linear model or multivariate logistic regression model was used to determine whether sleep quality affected the metabolic syndrome-related variables and the prevalence of MetS and its components after adjusting for depression and anxiety levels. In this study, it was demonstrated that increased waist circumference and central adiposity were present in more poor-quality sleepers than good-quality sleepers among women who experienced PMS. However, these results according to sleep quality were not shown among the women who experienced PMS. As noted in our results, women with PMS showed more psychological disorders. These negative moods might result in abnormal dietary behaviors, such as premenstrual food craving, emotional eating and appetite for specific tastes or foods [28], which could result in obesity, especially central adiposity [15,28]. Additionally, as mentioned in our previous study, sleep patterns were associated with increased dietary consumption and changes in hormone levels, which could lead to obesity [29]. Recent studies have shown that sleep quality is a crucial determinant of the development of MetS [30,31]. It is well known that high blood pressure or triglycerides are associated with adiposity, depression or anxiety and sleep disorders. Our results are similar to those of previous studies [16,32,33] that reported an association of premenstrual syndrome with blood pressure and suggested that premenstrual syndrome might be related to the future subsequent development of high blood pressure. In particular, increased diastolic blood pressure, as seen in our results, has been recognized as a powerful predictor of future risk of cardiovascular disease among women under the age of 50. An increased prevalence of triglyceridemia linked to sleep quality was observed among women with PMS in our study. As expected, sleep quality and PMS were associated with increased triglyceridemia. However, to explain the exact mechanism, future studies will be needed to analyze the combined effect of sleep quality and PMS on increased triglyceridemia.

According to several studies [6,7,8,9,10] on the relationship between PMS and dietary factors, dietary management is important as a modifiable parameter to prevent the presence of PMS or alleviate the severity of PMS. Additionally, dietary patterns that increase abdominal adiposity, such as higher consumption of calories or foods containing simple sugars, are well recognized to be associated with inadequate sleep patterns [29,34]. Although this study showed no significant differences in dietary nutrient density according to sleep quality regardless of the presence of PMS, the consumption of food groups was shown according to be linked to sleep quality and the presence of PMS. However, among subjects who experienced premenstrual syndrome, the consumption of alcohol in poor sleepers was 2.44 times higher than that in good sleepers in our study. Heavy consumption of alcohol might increase the burden of sleep quality as well as symptoms of PMS. Alcohol use was suggested to trigger negative moods, such as depression, anxiety and abnormal metabolic variables. In contrast, the consumption of eggs in poor sleepers was 1.75 times lower than that in good sleepers among subjects who experienced PMS. Additionally, the consumption of milk and dairy products in poor sleepers was 1.75 times lower than that in good sleepers (2.04 times lower) among subjects who did not experience PMS. For the dietary management of PMS, excessive consumption of total carbohydrates, total saturated fat and sodium and poor consumption of calcium, magnesium, zinc, thiamine, riboflavin, vitamin B6, vitamin D and vitamin E have been reported [6,7,8,9,10]. Eggs, milk and dairy products provide protein of high biological value, unsaturated fatty acids, a variety of minerals (calcium, phosphorus, potassium and choline) and fat-soluble and B vitamins [35,36]. The intake of these foods was suggested to have a potentially protective effect on MetS-related variables. However, the findings of our study on the effect of specific foods and nutrients on PMS could not be concluded due to insufficient scientific evidence, in addition to that of previous studies [37]. PMS was influenced by a complex of various factors other than food or nutrition. Therefore, self-care methods for PMS management and personalized nutrition treatment by specialists should be provided to reduce the severity of PMS [37]. Our cross-sectional study results should be interpreted cautiously because they cannot be used to identify causal relationships and can only be used for estimated associations. A limitation of our study is that the subjects were Korean women, so our findings are difficult to generalize to women from other countries. In addition, the presence of PMS was assessed using a self-report questionnaire, thereby making it difficult to identify the severity of PMS.

## 5. Conclusions

Using 307 premenopausal women in a cross-sectional study, our study demonstrated that the presence of PMS is related to different negative moods, such as depression and anxiety, and sleep quality. Poor sleep quality was associated with abnormal MetS-related variables and undesirable dietary consumption in women who experienced PMS but not in those who did not experience PMS. These findings suggest that the risk of MetS-related variables in Korean women experiencing PMS is associated with sleep quality, and these associations are likely modulated by psychological management and dietary control.

## Figures and Tables

**Figure 1 healthcare-11-01492-f001:**
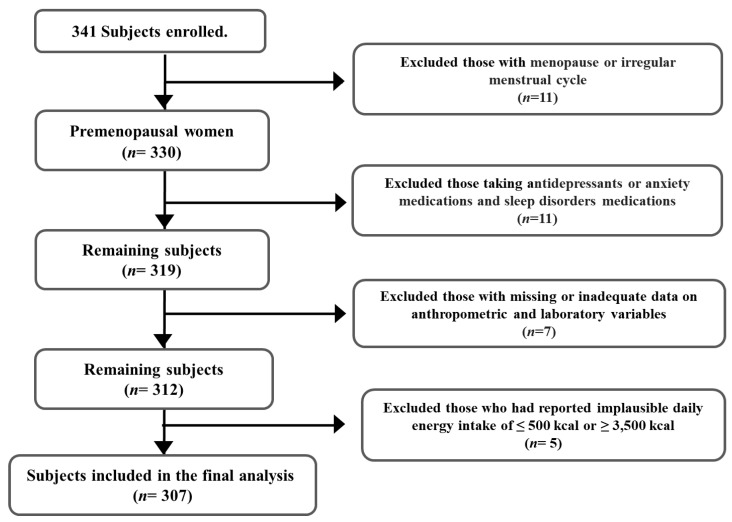
Subject selection.

**Figure 2 healthcare-11-01492-f002:**
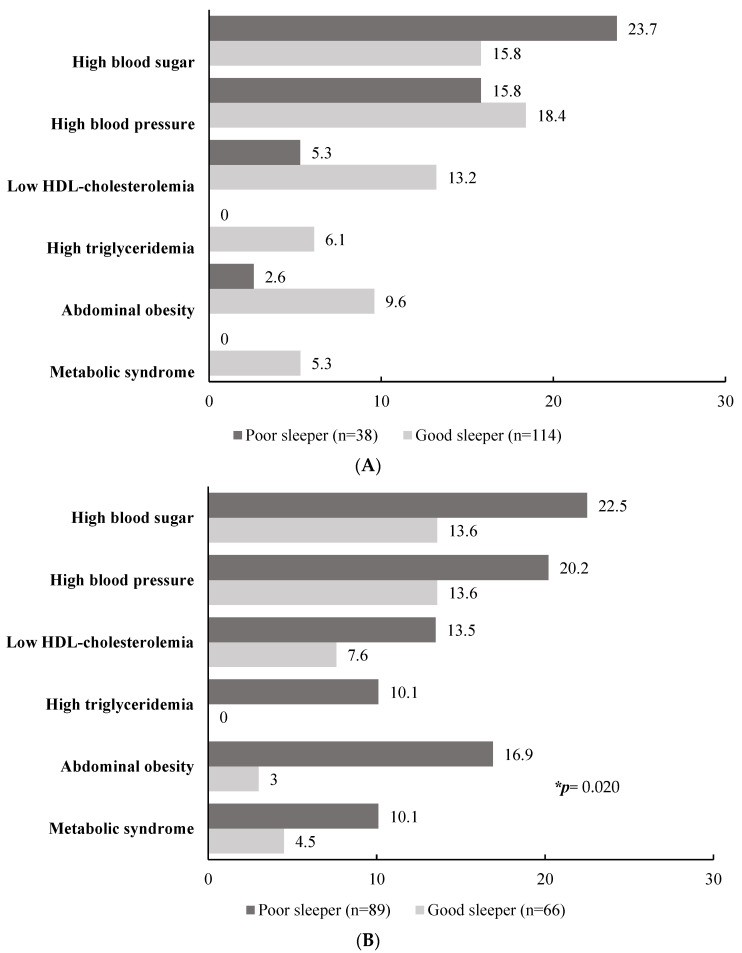
Prevalence of metabolic syndrome and its components by sleep quality and presence of premenstrual syndrome. (**A**) Did not experience; (**B**) experienced. The data are expressed as %. * Differences in sleep quality and the presence of premenstrual syndrome were analyzed using multivariate logistic regression, adjusting for depression level and anxiety level.

**Table 1 healthcare-11-01492-t001:** General characteristics by presence of premenstrual syndrome.

	Did Not Experience (n = 152)	Experienced (n = 155)	*p* Value *
Age, years	39.84 ± 5.87 (22–54)	39.31 ± 6.94 (21–53)	0.468
Educational level, ≥college	90.1	85.8	0.294
Occupation, housewife	24.3	32.3	0.131
Marriage status, yes	80.9	80.6	0.533
Current smoking, no	94.1	91.9	0.505
Alcohol drinking, no	35.5	29.0	0.272
Physical activity by IPAQ, low	46.1	44.5	0.566
Dietary supplementary intake, yes	32.2	26.5	0.316
Sleep quality by PSQI	4.45 ± 2.78	6.24 ± 2.82	<0.001
Depression level	28.54 ± 5.98	34.88 ± 8.40	<0.001
Anxiety level	35.57 ± 7.99	41.70 ± 8.52	<0.001

The data are expressed as the mean ± SD or %; * Differences in the presence of premenstrual syndrome were analyzed using the χ^2^ test, Fisher’s exact test or the *t* test. Abbreviations: IPAQ, International Physical Activity Questionnaire; PSQI, Pittsburgh Sleep Quality Index.

**Table 2 healthcare-11-01492-t002:** Metabolic syndrome-related variables by sleep quality and presence of premenstrual syndrome.

	Did Not Experience			Experienced		
	Good Sleeper (*n* = 114)	Poor Sleeper (*n* = 38)	*p* Value *	Good Sleeper (*n* = 66)	Poor Sleeper (*n* = 89)	*p* Value *
WC (cm)	72.38 ± 8.33	70.66 ± 5.42	0.372	70.36 ± 10.59	74.10 ± 9.88	0.018
SBP (mmHg)	114.63 ± 14.42	114.84 ± 12.51	0.552	112.55 ± 12.62	115.16 ± 14.84	0.258
DBP (mmHg)	69.69 ± 10.62	71.63 ± 9.79	0.559	67.95 ± 9.12	71.99 ± 10.93	0.012
FG (mg/dL)	94.18 ± 19.36	95.92 ± 7.66	0.611	90.64 ± 8.93	92.93 ± 10.10	0.129
TG (mg/dL)	88.43 ± 60.43	74.92 ± 20.71	0.398	78.58 ± 29.23	90.51 ± 46.98	0.169
HDL-C (mg/dL)	68.21 ± 15.75	75.55 ± 15.65	0.090	70.51 ± 16.88	69.23 ± 18.48	0.891

The data are expressed as the mean ± 95% CI; * Differences in sleep quality and presence of premenstrual syndrome were analyzed using a general linear model after adjusting for depression level and anxiety level. Abbreviations: WC, waist circumference; SBP, systolic blood pressure; DBP, diastolic blood pressure; FG, fasting glucose; TG, triglyceride; HDL-C, high-density lipoprotein cholesterol.

**Table 3 healthcare-11-01492-t003:** Dietary nutrient density by sleep quality and presence of premenstrual syndrome.

	Did Not Experience			Experienced		
	Good Sleeper (*n* = 114)	Poor Sleeper (*n* = 38)	*p* Value *	Good Sleeper (*n* = 66)	Poor Sleeper (*n* = 89)	*p* Value *
Carbohydrate (g)	120.67 ± 23.78	114.02 ± 28.18	0.131	119.98 ± 25.25	117.75 ± 29.58	0.789
Fat (g)	34.70 ± 9.30	35.91 ± 10.12	0.376	35.81 ± 9.28	34.69 ± 9.80	0.383
Protein (g)	43.94 ± 11.83	44.00 ± 11.16	0.881	42.72 ± 8.54	43.07 ± 10.28	0.795
Calcium (mg)	272.56 ± 112.60	249.30 ± 123.51	0.057	259.79 ± 108.71	260.22 ± 121.35	0.714
Phosphorus(mg)	651.50 ± 174.15	617.03 ± 160.97	0.209	631.15 ± 130.75	631.96 ± 160.83	0.755
Iron (mg)	8.53 ± 3.41	8.42 ± 4.19	0.721	7.82 ± 2.42	8.47 ± 3.71	0.117
Vitamin A (μg RAE)	253.63 ± 146.56	245.09 ± 126.84	0.458	256.44 ± 133.94	239.88 ± 139.44	0.714
Thiamin (mg)	1.18 ± 0.43	1.17 ± 0.45	0.863	1.17 ± 0.39	1.14 ± 0.45	0.431
Niacin (mg)	9.25 ± 4.76	8.77 ± 5.07	0.686	8.65 ± 3.28	8.20 ± 3.32	0.631
Folate (μg)	285.59 ± 122.31	303.21 ± 197.16	0.977	310.10 ± 149.60	278.98 ± 116.95	0.178
Vitamin C (mg)	63.85 ± 61.77	67.25 ± 84.43	0.989	64.51 ± 54.68	53.49 ± 45.96	0.322

The data are expressed as the mean ± SD; Dietary nutrient density (intake/1000 kcal); * Differences in sleep quality and presence of premenstrual syndrome were analyzed with a general linear model after adjusting for depression level and anxiety level.

**Table 4 healthcare-11-01492-t004:** Food groups per 1000 kcal by sleep quality and presence of premenstrual syndrome.

	Did Not Experience			Experienced		
	Good Sleeper (*n* = 114)	Poor Sleeper (*n* = 38)	*p* Value *	Good Sleeper (*n* = 66)	Poor Sleeper (*n* = 89)	*p* Value *
Grains (g)	110.76 ± 49.07 104.50 (84.12–130.50)	97.37 ± 41.02 89.61 (70.23–128.69)	0.133	122.04 ± 56.52 113.64 (84.90–145.52)	108.33 ± 51.73 95.60 (76.89–129.91)	0.076
Sugar (g)	5.32 ± 6.44 2.86 (1.19–7.02)	4.11 ± 4.17 3.46 (0.77–6.21)	0.152	4.00 ± 4.09 2.92 (0.90–5.69)	4.77 ± 5.22 3.40 (1.46–6.30)	0.214
Soy and products (g)	17.14 ± 33.25 4.61 (0.00–19.28)	9.89 ± 22.82 3.18 (0.00–9.08)	0.431	20.89 ± 37.89 4.98 (0.00–25.54)	16.40 ± 35.31 2.59 (0.00–16.67)	0.449
Vegetables and mushrooms (g)	159.47 ± 97.38 142.44 (100.43–201.37)	158.24 ± 71.92 160.25 (98.17–212.14)	0.494	166.12 ± 108.77 153.84 (106.64–210.84)	156.24 ± 72.16 148.50 (102.01–206.58)	0.505
Fruits and products (g)	110.30 ± 105.47 92.84 (16.47–161.43)	122.83 ± 127.68 94.48 (4.44–203.33)	0.757	109.81 ± 116.13 80.19 (0.00–188.10)	110.70 ± 116.09 90.06 (0.00–163.82)	0.724
Meat (g)	78.37 ± 52.70 73.26 (46.49–104.35)	94.93 ± 63.99 85.70 (45.56–137.53)	0.072	71.90 ± 39.30 72.37 (49.46–100.19)	75.72 ± 44.99 74.30 (43.67–107.46)	0.787
Eggs (g)	14.03 ± 21.84 0.00 (0.00–24.60)	18.90 ± 22.36 10.41 (0.00–31.79)	0.466	29.39 ± 39.12 0.00 (0.00–43.71)	16.69 ± 28.14 0.00 (0.00–30.32)	0.028
Fish and shellfish (g)	37.57 ± 50.43 9.06 (0.00–60.91)	40.35 ± 56.33 10.01 (0.00–65.30)	0.950	32.44 ± 38.92 14.54 (0.00–60.72)	30.60 ± 40.41 9.18 (0.00–51.81)	0.773
Milk and dairy products(g)	57.56 ± 75.98 0.00 (0.00–112.19)	28.84 ± 48.27 0.00 (0.00–53.97)	0.020	43.16 ± 73.77 0.00 (0.00–80.21)	54.73 ± 91.18 0.00 (0.00–94.59)	0.345
Oil (g)	4.61 ± 3.33 4.07 (2.10–6.18)	4.11 ± 3.10 3.58 (1.98–5.81)	0.183	4.63 ± 3.58 3.98 (1.97–5.83)	4.05 ± 3.13 3.56 (1.46–6.21)	0.403
Seasoning (g)	15.26 ± 9.02 15.06 (7.89–20.07)	15.40 ± 7.77 15.11 (9.07–20.13)	0.817	13.99 ± 9.35 11.40 (7.74–19.99)	14.57 ± 8.91 13.58 (8.68–17.76)	0.663
Alcoholic drinks (g)	40.91 ± 91.66 1.34 (0.00–37.40)	40.51 ± 110.34 0.15 (0.00–41.52)	0.914	24.90 ± 42.00 0.00 (0.00–34.88)	70.19 ± 113.87 25.75 (0.00–106.55)	0.006

The data are expressed as the mean ± SD or median (interquartile range 25th–75th percentile); * Differences in sleep quality and the presence of premenstrual syndrome were analyzed with a general linear model after adjusting for depression level and anxiety level.

## Data Availability

The data used in this study are not available to the public due to ethical considerations.

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
