# Peer review of "Sleep Quality in Women with Premenstrual Syndrome Is Associated with Metabolic Syndrome-Related Variables"

_healthcare, 2023, doi:10.3390/healthcare11101492_

Round 1
Reviewer 1 Report
Reviewer Comments
In this research article, authors examined whether metabolic syndrome (MetS)-related variables are simultaneously affected by sleep quality and premenstrual syndrome (PMS) and by dietary consumption and found that women experiencing PMS had significantly lower sleep quality and were more depressed and anxious. The article concept is good but the selection criteria are not very clear. The result outcomes of this present paper are not very well correlated and discussed with various available recent research papers. Hence a thorough revision should be needed to improve the manuscript based on the below comments.
Major comment
1. Add more specific content with proper citations in the discussion section to enhance the quality of the paper.
2. The selection criteria of the subjects and various parameters undertaken for the study should be added in tabular format for better understanding.
3. Graphical representation of the resulting data should be incorporated for easy understanding for the readers.
4. References should be provided exactly in the journal’s specific format. Please check once and rewrite (reference no. 15, 17, and more).
5. Framing of sentences is very poor, especially in the result and discussion section.
Minor comments
- Add recent references that support your study and resulting outcomes to enhance and enlighten the quality of your paper.
- Lines 31-33, need a specific reference.
- Mention the age criteria (min & max-age) for premenopausal women used for the cross-sectional study because age is a major interfering factor for metabolism and sleep.
- Line 68, “treatment of sleep disorders” is inappropriate. Reconstruct the sentence.
- Line 94, “The Center for Epidemiological Studies-Depression”, T of The should be small.
- In Line 205, 319 will be replaced by 293. In the method and conclusion, you mentioned 293 actual participants for this study.
- Discuss and compare your results with the following paper: DOI: 10.1080/09513590.2021.1968820
DOI: 10.1055/a-1519-7517
DOI: 10.1007/s00737-019-00984-2
8. Discuss the following papers in the discussion section:
doi: 10.3390/healthcare8020186
doi: https://doi.org/10.1016/j.psychres.2019.01.096
doi: https://doi.org/10.3389/fnut.2023.1079417
Author Response
In this research article, authors examined whether metabolic syndrome (MetS)-related variables are simultaneously affected by sleep quality and premenstrual syndrome (PMS) and by dietary consumption and found that women experiencing PMS had significantly lower sleep quality and were more depressed and anxious. The article concept is good but the selection criteria are not very clear. The result outcomes of this present paper are not very well correlated and discussed with various available recent research papers. Hence a thorough revision should be needed to improve the manuscript based on the below comments.
We appreciated the reviewer for careful reading and description about our manuscript with the valuable comments. We worked to the best of our abilities to revise the issues reviewer point out.
- Add more specific content with proper citations in the discussion section to enhance the quality of the paper.
We appreciate the constructive and very helpful comments.
It has been modified based on the comment as follows.
“Meers et al. reported that sleep deprivation and psychological abnormalities co-occurred, rather than existing as separate correlated factors associated with the severity of PMS and negative mood [26]”
“Recent studies have shown that sleep quality is a crucial determinant of the development of MetS [30, 31].”
- The selection criteria of the subjects and various parameters undertaken for the study should be added in tabular format for better understanding.
We appreciate the constructive and very helpful comments.
It has been modified based on the comment as follows.
“Before participating in the survey, among women who wanted to participate voluntarily, we asked if they met the following inclusion criteria: 1) having a regular menstrual cycle; 2) not having pregnant or breastfeeding status; 3) not having depression or anxiety; and 4) not having sleep disorders (Fig. 1). A total of 341 subjects satisfied the criteria and were included. We conducted a comprehensive face-to-face interview and assessed the anthropometric and laboratory variables of the subjects. Through subsequent hormone tests, we excluded subjects who experienced menopause or irregular menstrual cycles due to polycystic ovary syndrome or prolactinoma (n=11). Additionally, we excluded women taking antidepressants or anxiety medications (n=4) and those taking sleep disorder medications (n=7). Furthermore, subjects with missing or inadequate anthropometric and laboratory data for assessing MetS (n=7) and those having implausible daily consumption (total energy consumption of ≤500 kcal or ≥3,500 kcal) were excluded (n=5). Finally, a total of 307 participants were included in the analyses.”
- Graphical representation of the resulting data should be incorporated for easy understanding for the readers.
As you suggested, Table 3 has been changed Fig2.
- References should be provided exactly in the journal’s specific format. Please check once and rewrite (reference no. 15, 17, and more).
It has been modified based on the comments.
- Framing of sentences is very poor, especially in the result and discussion section.
We worked to best of our abilities to improve the results and discussion sections.
Minor comments
- Add recent references that support your study and resulting outcomes to enhance and enlighten the quality of your paper.
As you suggested, recent references were added for supporting and enhancing our results.
- Lines 31-33, need a specific reference.
It has been added based on the comments.
- Mention the age criteria (min & max-age) for premenopausal women used for the cross-sectional study because age is a major interfering factor for metabolism and sleep.
It has been added based on the comments in Table 1.
- Line 68, “treatment of sleep disorders” is inappropriate. Reconstruct the sentence.
It has been modified based on the comments as follow
- Line 94, “The Center for Epidemiological Studies-Depression”, T of The should be small.
It has been modified based on the comments
- In Line 205, 319 will be replaced by 293. In the method and conclusion, you mentioned 293 actual participants for this study.
It has been modified based on the comments.
- Discuss and compare your results with the following paper:
DOI: 10.1080/09513590.2021.1968820
DOI: 10.1055/a-1519-7517
DOI: 10.1007/s00737-019-00984-2
It has been modified based on the comments.
- Discuss the following papers in the discussion section:
doi: 10.3390/healthcare8020186
doi: https://doi.org/10.1016/j.psychres.2019.01.096
doi: https://doi.org/10.3389/fnut.2023.1079417
It has been modified based on the comments.
Reviewer 2 Report
Review for the manuscript entitled “Sleep Quality in Women with Premenstrual Syndrome is Associated with Metabolic Syndrome-Related Variables”.
This cross-sectional study suggests that Korean women who experience PMS with sleep disturbance may have a higher prevalence of MetS components. In addition, idiosyncratic dietary habits may be associated.
Although the conclusions are valid, the reviewer has several concerns.
1. Additional data or discussion on exercise habits are needed.
2. The statistical analysis methods chosen were difficult to understand. The reviewer thinks as follows: Fischer's exact test or Mann-Whitney U test in Table 1, Mann-Whitney U test in Table 2&4&5, Fischer's exact test in Table 3 would be commonly used. Furthermore, the significance of always correcting for depression and anxiety levels was unclear.
3. At least factors that show Mean<SD are not considered parametric. In such cases they are usually expressed as median (IQR).
4. The authors conclude: "These associations may be modified by psychological management and dietary control".
However, as this is a cross-sectional study, the limitation is that the causal relationship is not clear.
That is all.
Author Response
This cross-sectional study suggests that Korean women who experience PMS with sleep disturbance may have a higher prevalence of MetS components. In addition, idiosyncratic dietary habits may be associated. Although the conclusions are valid, the reviewer has several concerns.
We appreciated the reviewer for careful reading and description about our manuscript with the valuable comments. We worked to the best of our abilities to revise the issues reviewer point out.
- Additional data or discussion on exercise habits are needed.
We appreciate the constructive and very helpful comments.
We agreed that analysis of exercise habits of subjects are needed. The physical activity is an important lifestyles and management of MetS. So, we assessed physical activity using international physiology activity questionnaire (IPAQ). However, physical activity according to the presence of premenstrual syndrome were not significantly different as well as sleep quality. Therefore, future studies will be needed to confirmed effects premenstrual syndrome or sleep quality according to specific types of physical activity on MetS.
- The statistical analysis methods chosen were difficult to understand. The reviewer thinks as follows: Fischer's exact test or Mann-Whitney U test in Table 1, Mann-Whitney U test in Table 2&4&5, Fischer's exact test in Table 3 would be commonly used. Furthermore, the significance of always correcting for depression and anxiety levels was unclear.
Thank you for your comment.
It has been modified based on the comment as follows.
“Tests of normality for all continuous variables were carried out before the analyses, and skewed variables were logarithmically transformed. Data are presented as back-transformed means and 95% confidence intervals (CIs).”
However, sample size of our study for statical analysis is not small, it is not necessary to perform nonparmetric statistical analysis (Mann- Whiney U-test, and Fisher’s exact in Table 1). In our study, because of depression level and anxiety level as covariates, a general linear model was used in Table 2 ,4, and 5. Furthermore, Table 3 was used multivariate logistic regression due to depression level and anxiety level as covariates.
In the Table 5, Food groups consumption were greatly affected by those energy, dietary energy was additionally adjusted. But based on your comment, we modified the food group per 1000kcal was modified, and adjusted as covariate depression level and anxiety level.
“And Differences in food groups per 1000 kcal according to sleep quality and the presence of PMS were analyzed using a general linear model after adjusting for depression level and anxiety level”
- At least factors that show Mean<SD are not considered parametric. In such cases they are usually expressed as median (IQR).
It has been modified based on the comments.
- The authors conclude: "These associations may be modified by psychological management and dietary control".
However, as this is a cross-sectional study, the limitation is that the causal relationship is not clear.
It has been modified based on the comments as follows.
“These findings suggest that the risk of MetS-related variables in Korean women experiencing PMS is associated with sleep quality, and those associations are likely modulated by psychological management and dietary control.”
Reviewer 3 Report
Thank you for the opportunity to review this interesting manuscript. The authors investigated the relationship between sleep quality, premenstrual syndrome, and metabolic syndrome criteria. The authors have conducted an interesting study and the results provide relevant information about the associations between these variables.
The English language of the manuscript needs to be revised for clarity and readability. Certain sentence structures and terms used create confusion (e.g., L34-35 about sleep problems affecting insomnia; l. 219-220 about not observing MetS symptoms; or experienced vs. inexperienced in Table 1). Please consider using professional editing services to enhance the language and improve reader understanding.
L. 254 Avoid unwarranted suggestions of causality. From this type of design, It is not possible to say whether the participants’ alcohol consumption impaired their sleeping or participants consumed alcohol to help their existing sleeping issues.
L. 268 It is somewhat inappropriate to talk about ‘using’ participants. Kindly consider rewording.
L. 273-274 The las sentence of the conclusion is not warranted. Based on the current study, it is not possible to say that psychological management and dietary control might modify the association between MetS and sleep quality in those with PMS.
L. 274 The term dietary control is somewhat inappropriate as it suggests restrictions. Kindly consider using ‘dietary management’ instead.
Minor comment: MetS is spelled inconsistently throughout the manuscript. Please ensure that a consistent abbreviation is used.
Author Response
Thank you for the opportunity to review this interesting manuscript. The authors investigated the relationship between sleep quality, premenstrual syndrome, and metabolic syndrome criteria. The authors have conducted an interesting study and the results provide relevant information about the associations between these variables.
We appreciated the reviewer for careful reading and description about our manuscript with the valuable comments. We worked to the best of our abilities to revise the issues reviewer point out.
The English language of the manuscript needs to be revised for clarity and readability. Certain sentence structures and terms used create confusion (e.g., L34-35 about sleep problems affecting insomnia; l. 219-220 about not observing MetS symptoms; or experienced vs. inexperienced in Table 1). Please consider using professional editing services to enhance the language and improve reader understanding.
Thank you for your comment.
We agree to your comment that our manuscript needs to be English editing services.
- 254 Avoid unwarranted suggestions of causality. From this type of design, It is not possible to say whether the participants’ alcohol consumption impaired their sleeping or participants consumed alcohol to help their existing sleeping issues.
We appreciate the constructive and very helpful comments.
It has been modified based on the comments and has been added the limitations.
“. Heavy consumption of alcohol might increase the burden of sleep quality as well as symptoms of PMS.”
“However, the findings of our study on the effect of specific foods and nutrients on PMS could not be concluded due to insufficient scientific evidence in addition to that of previous studies [37]. PMS was influenced by a complex of various factors other than food or nutrition. Therefore, self-care methods for PMS management and personalized nutrition treatment by specialists should be provided to reduce the severity of PMS [37]. Our cross-sectional study results should be interpreted cautiously because they cannot be used to identify causal relationships and can only be used for estimated associations. A limitation of our study is that the subjects were Korean women, so our findings are difficult to generalize to women from other countries. In addition, the presence of PMS was assessed using a self-report questionnaire, thereby making it difficult to identify the severity of PMS”.
- 268 It is somewhat inappropriate to talk about ‘using’ participants. Kindly consider rewording.
Our manuscript was reviewed English editing services.
- 273-274 The las sentence of the conclusion is not warranted. Based on the current study, it is not possible to say that psychological management and dietary control might modify the association between MetS and sleep quality in those with PMS.
It has been modified based on the comments as follows.
“These findings suggest that the risk of MetS-related variables in Korean women experiencing PMS is associated with sleep quality, and those associations are likely modulated by psychological management and dietary control.”
- 274 The term dietary control is somewhat inappropriate as it suggests restrictions. Kindly consider using ‘dietary management’ instead.
Our manuscript was reviewed English editing services.
Minor comment: MetS is spelled inconsistently throughout the manuscript. Please ensure that a consistent abbreviation is used.
It has been modified based on the comments throughout the manuscript.
Reviewer 4 Report
This study was conducted to examine the difference in metabolic syndrome factors according to sleep quality in women who experienced premenstral syndrome. Considering that both sleep disorders and premenstrual syndrome are factors that can degrade the quality of life of women, this study is of considerable value. However, it is necessary to confirm and correct the following matters.
1. Please provide a more detailed method of recruiting subjects.
2. It is necessary to check the number of subjects.
- The total number of study subjects is 293. However, PMS (n=159) and unexperienced PMS (n=153) have more than 300 people. In addition, the number of people analyzed for metabolic syndrome factor results is smaller than that (Table 3), and the total number of people analyzed for the meal intake survey results is 314.
3. A full review of the words used is required.
- For example, dietary intake is more appropriate than dietary consumption.
4. Alcohol intake was significantly higher in the poor sleeper group than in the good sleeper group in the group that experienced PMS. However, as a result of a one-day dietary recall, an additional proportion test for usual drinking is needed.
5. The reference has been pushed back from number 15. Please revise the whole thing.
Author Response
This study was conducted to examine the difference in metabolic syndrome factors according to sleep quality in women who experienced premenstral syndrome. Considering that both sleep disorders and premenstrual syndrome are factors that can degrade the quality of life of women, this study is of considerable value. However, it is necessary to confirm and correct the following matters.
We appreciated the reviewer for careful reading and description about our manuscript with the valuable comments. We worked to the best of our abilities to revise the issues reviewer point out.
- Please provide a more detailed method of recruiting subjects.
We appreciate the constructive and very helpful comments.
It has been modified based on the comment as follows.
“Before participating in the survey, among women who wanted to participate voluntarily, we asked if they met the following inclusion criteria: 1) having a regular menstrual cycle; 2) not having pregnant or breastfeeding status; 3) not having depression or anxiety; and 4) not having sleep disorders (Fig. 1). A total of 341 subjects satisfied the criteria and were included. We conducted a comprehensive face-to-face interview and assessed the anthropometric and laboratory variables of the subjects. Through subsequent hormone tests, we excluded subjects who experienced menopause or irregular menstrual cycles due to polycystic ovary syndrome or prolactinoma (n=11). Additionally, we excluded women taking antidepressants or anxiety medications (n=4) and those taking sleep disorder medications (n=7). Furthermore, subjects with missing or inadequate anthropometric and laboratory data for assessing MetS (n=7) and those having implausible daily consumption (total energy consumption of ≤500 kcal or ≥3,500 kcal) were excluded (n=5). Finally, a total of 307 participants were included in the analyses.”
- It is necessary to check the number of subjects.
- The total number of study subjects is 293. However, PMS (n=159) and unexperienced PMS (n=153) have more than 300 people. In addition, the number of people analyzed for metabolic syndrome factor results is smaller than that (Table 3), and the total number of people analyzed for the meal intake survey results is 314.
- The total number of study subjects is 293. However, PMS (n=159) and unexperienced PMS (n=153) have more than 300 people. In addition, the number of people analyzed for metabolic syndrome factor results is smaller than that (Table 3), and the total number of people analyzed for the meal intake survey results is 314.
Thank you for your comment.
We checked the number of subjects all throughout the manuscript.
- A full review of the words used is required.
- For example, dietary intake is more appropriate than dietary consumption.
Our manuscript was reviewed English editing services.
- Alcohol intake was significantly higher in the poor sleeper group than in the good sleeper group in the group that experienced PMS. However, as a result of a one-day dietary recall, an additional proportion test for usual drinking is needed.
Thank you for your comment.
Although dietary consumption was investigated using single day dietary recall methods, usual amount alcohol consumed was investigated because subjects who visit the hospital for health checkup in general, do not drink alcohol the day before.
- The reference has been pushed back from number 15. Please revise the whole thing.
It has been modified based on the comments.
Reviewer 5 Report
This manuscript presents a cross-sectional study identifying women with and without the premenstrual syndrome, in which the authors detect the frequencies and eventual associations between sleep quality and MetS-related variables.
The results answer the question correctly; however, the presentation of the data implies a longitudinal design as the authors repeatedly use the notion of “the effect of sleep quality on MetS variables,” clearly establishing a line time and suggesting a cause-effect proving that the present study is not supporting.
The manuscript should be restructured to clarify the association level without suggesting an etiological role for any analyzed variable.
Author Response
We appreciated the reviewer for careful reading and description about our manuscript with the valuable comments.
We worked to the best of our abilities to revise restructured to clarify the association level without suggesting an etiological role for any analyzed variable.

Round 2
Reviewer 2 Report
The authors appear to have responded as adequately as possible to the previous comments.
Reviewer 4 Report
The authors have adequately addressed all comments with corresponding revisions. I have no further comments.